# Association of HIV infection and antiretroviral therapy with the occurrence of an unfavorable TB treatment outcome in a rural district hospital in Eastern Cape, South Africa: A retrospective cohort study

**Brittney J. van de Water**[1]*, **Isabel Fulcher**[2], **Suretha Cilliers**[3], **Nadishani Meyer**[3], **Michael Wilson**[4], **Catherine Young**[5], **Ben Gaunt**[3,6], **Karl le Roux**[3,6,7]

1 Connell School of Nursing, Boston College, Chestnut Hill, MA, United States of America, 2 Department of Global Health and Social Medicine, Harvard Medical School, Boston, MA, United States of America, 3 Zithulele District Hospital, Mqanduli, Eastern Cape, South Africa, 4 Advance Access & Delivery, Durban, Kwa-Zulu Natal, South Africa, 5 Jabulani Rural Health Foundation, Mqanduli, Eastern Cape, South Africa, 6 Family Medicine Department, Walter Sisulu University, Mthatha, Eastern Cape, South Africa, 7 Primary Health Care Directorate, University of Cape Town, Cape Town, Western Cape, South Africa

* Brittney.vandewater@bc.edu

**Data Availability Statement:** The data underlying the results presented in the study are available

## Abstract

### Background

Our objective was to assess differences in TB treatment outcomes between individuals who were HIV negative, HIV positive on anti-retroviral treatment (ART) and HIV positive not on ART, at TB treatment initiation at a rural district hospital in Eastern Cape, South Africa.

### Methods

This was a retrospective cohort study of individuals diagnosed with TB between January 2017 and April 2020 at a district hospital. Adults 15 years and over with reported HIV status and treatment outcome were included (N = 711). A categorical outcome with three levels was considered: unfavorable, down referral, and success. We report descriptive statistics for the association between HIV and ART status and treatment outcome using Chi-square and Fisher's exact tests. A multinomial baseline logit model was used to estimate odds ratios for treatment outcomes.

### Results

Overall, 59% of included patients were HIV positive with 75% on ART. Eighty-eight patients 12% had an unfavorable outcome. Half of all patients were down referred with an additional 37% having a successful outcome. Individuals without HIV were more likely to be down referred (versus unfavorable) compared to individuals with untreated HIV (2.90 OR, 1.36, 6.17 95% CI). There was a greater likelihood for individuals without HIV having a successful

from the Boston College DavaVerse Repository: https://doi.org/10.7910/DVN/NQYR7W.

**Funding:** This work was supported by the Robert Wood Johnson Foundation Future of Nursing Scholars post-doctoral program [74652 to BvdW] and National Institute of Nursing Research at the National Institutes of Health [1K23NR019019-01A1 to BvdW].

**Competing interests:** The authors have declared that no competing interests exist.

TB treatment outcome compared to individuals with untreated HIV (4.98 OR, 2.07, 11.25 95% CI).

## Conclusion

The majority of individuals had positive TB treatment outcomes (down referred or success). However, people without HIV had nearly five times greater odds of having successful outcomes than those with untreated HIV.

## Background

Tuberculosis (TB) is the leading cause of infectious disease death worldwide, and in South Africa (WHO, 2019) [1]. Nearly one in every four deaths among people living with HIV is attributable to TB [1]. South Africa has an HIV prevalence rate of 20.4% among adults, 8.7% among youth 15 to 24 years, and 2.7% among children < 2 years and has more cases of HIV than any other country, with 7.7 million known HIV infections in 2018 [2, 3]. In a recent country-wide prevalence survey, it is estimated that the prevalence of bacteriologically confirmed pulmonary TB in South Africa is 852 (95% 679–1,026) per 100,000 population among individuals 15 years and older [4]. Therefore, South Africa has a dual burden of both HIV and TB, with nearly 60% of individuals diagnosed with TB disease being co-infected with HIV [3]. Yet, despite the incredibly high prevalence of HIV in South Africa, TB (and TB/HIV) continues to kill more South Africans annually than HIV alone [1, 5].

TB disease is the most serious and common opportunistic infection affecting people living with HIV [6–8]. However, since the era of antiretroviral therapy (ART), individuals dually infected with HIV and TB have had increased survival rates [6, 9, 10]. There is currently a lack of evidence regarding HIV status, ART status (or suppressed viral loads) and TB treatment outcomes [11–13]. However, ART is known to modify the effect of HIV on TB treatment outcomes [14]. A systematic review and meta-analysis of early ART initiation among patients with TB found that early ART initiation during TB treatment was associated with reduced all-cause mortality; however, it also had an increased rate of TB-associated immune reconstitution inflammatory syndrome (IRIS) and death related to TB-IRIS especially if started within the first two weeks of initiating TB treatment [6, 15, 16].

The advent of ART has changed the landscape of HIV [17]. However, among people living with HIV, the risk of developing TB is still 26 times higher than the general population [1]. Challenges for co-treatment remain; however, recent literature describes advances in effective drugs, more sensitive and rapid molecular testing for TB, and use of shorter regimens have improved outcomes among individuals co-infected with TB and HIV [18].

The objective of this study was to assess the relationship between HIV and ART status with TB treatment outcomes among a cohort diagnosed with TB in a rural, high HIV prevalence setting.

## Methods

### Study design and study population

Data for this study were taken from TB medical charts of a cohort of individuals 15 years or older who were diagnosed with drug-sensitive TB and who should have completed treatment at the district hospital between January 4, 2017 and April 27, 2020 (N = 1006) (Fig 1). There were 295

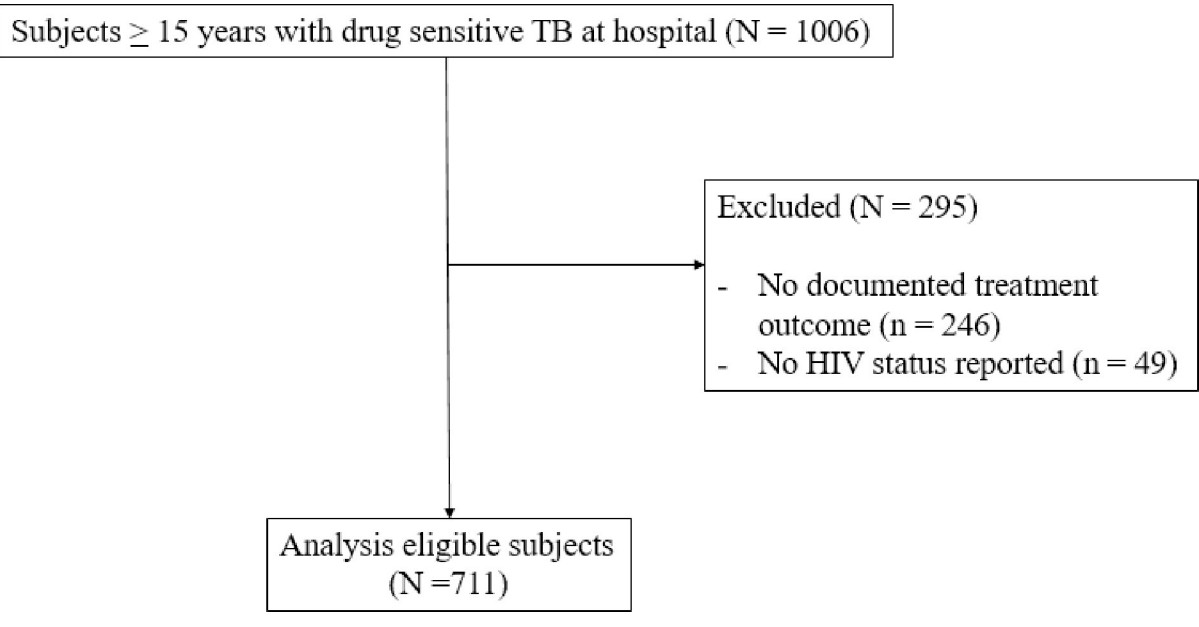

**Fig 1. Population flowchart.**

individuals excluded due to no documented treatment outcome (n = 246) or no HIV status reported (n = 49) for a final sample size of 711. We excluded children < 15 years due to United Nations age group definitions, differing treatment guidelines and outcomes among children, and because individuals < 15 years often are seen in pediatric settings [19–21]. Additionally, the vast majority of adults ≥15 years with drug sensitive TB in South Africa are managed with the same four drug regimen for the first two months of the "intensive phase" of treatment: rifampicin, isoniazid, pyrazinamide and ethambutol (RHZE), followed by four months of rifampicin and isoniazid (RH) for the "continuation phase" as per the South African TB guidelines [5].

## Setting

This study took place in the King Sabata Dalindyebo (KSD) sub-district of the OR Tambo District, rural Eastern Cape, South Africa. This is one of the poorest districts in the country and one of ten districts countrywide with the highest dual burden of HIV and TB [5, 22]. The Eastern Cape has a TB incidence of 839/100,000 in 2017, higher than the country prevalence estimated at 737/100,000 [4]. Additionally, the Eastern Cape has a high overall HIV prevalence rate (25.2%) as of 2017 [22–24]. This district hospital and its surrounding 12 clinics serve a deeply rural area of the Eastern Cape with a catchment population of approximately 127,500 people [24]. Access to clinics, and particularly to the district hospital, is challenging due to the hilly geography of the area, a lack of road infrastructure, and household financial constraints. As of the latest Census data from 2011, the annual median household income is R14600 (US $1000), with only 9.7% of adults having employment, 9.6% unemployed, 10.5% discouraged work seekers, and 70.2% who are not economically active [25].

## Data sources and study measures

Baseline demographic and medical history were collected through self-report and an interview with a healthcare provider as the TB file was started, GeneXpert® Ultra was performed on most patients and when required, drug-susceptibility testing (DST) results for first- and

second-line TB drugs were obtained through sputum collection sent to the South African National Health Laboratory Services (NHLS). TB disease was diagnosed by clinical examination, chest x-ray, GeneXpert® Ultra or sputum smear microscopy. HIV was self-reported or confirmed with two rapid cartridge HIV antibody tests, with ELISA blood test if discrepancy between rapid tests [26]. All data were entered into and downloaded from REDCap, an electronic data management system [27].

Treatment outcome definitions are in accordance with the WHO 2013 revised definitions and reporting framework for tuberculosis [28]. Composite categorical outcomes with three levels were created: unfavorable, down referral, and success. Unfavorable outcome was defined as loss to follow-up, death, or treatment failure. Loss to follow-up (LTFU) is defined as a TB patient who did not start treatment or whose treatment was interrupted for two consecutive months or more and thus considered an unfavorable outcome, aligning with existing TB literature [29, 30]. Success was defined as cure or completed treatment. Down referral refers to patients who are down referred to be treated at their nearest primary care clinic where we could no longer follow them. The outcomes for these patients were therefore uncertain; however, down-referred patients are typically stable and doing well on treatment, such that their clinicians believe they can successfully be treated at their local clinic. Down referral is strongly encouraged and ensures that patients are treated closer to their homes, enabling better adherence. It is seen as a positive outcome and helps TB care to be more patient-centered and decentralized [5, 31]. The exposure of interest was categorical HIV and ART status: HIV infection on ART at the initiation of TB treatment (treated HIV), HIV infection but not on ART at initiation of TB treatment (untreated HIV), and no HIV infection. Information on covariates was also collected from medical histories in the medical chart including: age, sex, smoking status, alcohol use, prior/current employment in a mine, and presence of comorbidities (hypertension, diabetes, epilepsy, mental illness, liver disease, and renal insufficiency). Missing values for covariates were given their own category of "Missing", often when a clinician did not ask or did not tick "yes" or "no" in the medical chart. For missing HIV status, often it was due to tests not being available on the day a patient was seen, and then no HIV status was recorded upon testing on a subsequent day after the initial baseline data collection. Finally, a final treatment outcome is not always recorded by clinicians in the medical chart. Therefore, we believe these missing data points are missing at random.

## Statistical analysis

We first report descriptive statistics for the association between HIV and ART status and the measured demographic and clinical covariates, including treatment outcome, using Wilcoxon signed rank and Fisher's exact tests. Demographic covariates include age, sex, past employment in a mine, tobacco use, and alcohol use. Clinical covariates include GeneXpert® Ultra for diagnosis, TB type, and multiple comorbidities.

We use a multinomial baseline logit model to estimate the odds ratios for treatment outcome (reference level: unfavorable) and HIV and ART status (reference level: no HIV infection). We adjust for age, sex, TB type, tobacco use, and alcohol use as these are known confounders of the relationship between HIV and ART status and TB treatment outcomes [22, 32, 33]. We categorized age as 15–24, 25–54, 55–65, and 65+ to capture the non-linear relationship between age and TB treatment outcomes and in standardization with other reports of TB treatment outcomes (Fig 2) [4, 34]. Missing values for TB type, alcohol, and tobacco use were made into their own "Unanswered" category.

To test our assumption that the treatment outcomes and HIV status were missing completely at random, we provide supplementary analyses (S1 File). First, we compared the

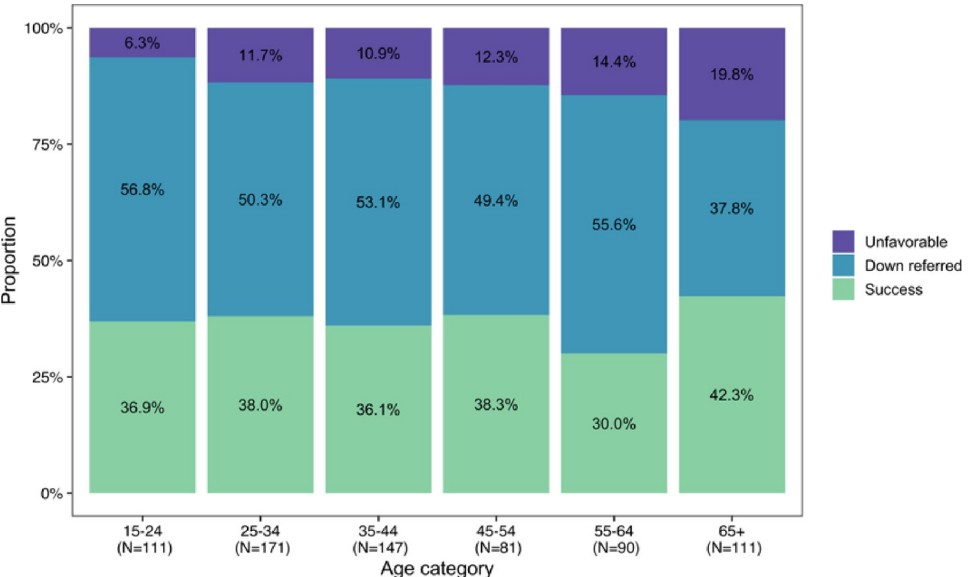

**Fig 2. TB treatment outcomes by age group (N = 936).**

distribution of demographic characteristics between those included and excluded from the main analysis. In addition, we used multiple imputation by chained equations to create 20 datasets with imputed values for missing variables and pool the results from the main analyses. This will provide alternate odds ratio estimates assuming the data are missing conditionally at random. R V3.6.0 was used for data cleaning and statistical analyses with the tidyverse, nnet, and mice software packages.

## Ethics approval

Ethics approval was provided by Harvard Medical School (IRB19-0605) and Walter Sisulu University (049/2019) as well as the Eastern Cape Department of Health (EC_201908_011). Data used in this retrospective study were identifiable upon collection; however, were fully deidentified and anonymized before statistical analysis. Both IRBs waived informed consent.

## Results

Overall, 418/711 (58.8%) of included patients were HIV positive with 75% of these patients on ART (Table 1). The median age of patients was 38 years (interquartile range 30–57) with those who were HIV negative being more likely to be older (p<0.001). Half (50.0%) of patients were male, with males less likely to be HIV positive and on ART compared to females, but also more likely to not be on ART if HIV positive (p<0.001). The majority (62.0%) of patients had pulmonary TB, while 30.0% had extra pulmonary TB, 4.4% had both pulmonary and extra pulmonary, and 3.7% had TB type not recorded. Most (70.2%) patients were diagnosed using GeneXpert® Ultra, while 66.8% had a chest x-ray to assist with or confirm diagnosis. The tobacco use, alcohol use, and prior or current mining work variables were unanswered (missing) for about one third of the cohort, though the rates of missing data did not vary significantly by HIV and ART status. Among those who answered these questions, 20.0% reported tobacco use, 22.9% reported alcohol use, and 9.9% reported mining work. HIV negative persons were more likely to have been involved in mining work (p<0.001); there were no differences by alcohol or tobacco use. Additionally, about one third of patients did not have

**Table 1. Demographics of patients diagnosed with TB at a district hospital based on HIV and ART status (N = 711).**

| | HIV+, no ART (N = 104) | HIV+, ART (N = 314) | HIV- (N = 293) | Total (N = 711) | P-value[1] |
|---|---|---|---|---|---|
| Age | | | | | <0.001 |
| Median (IQR) | 32 (28, 44) | 37 (31, 45) | 53 (28, 67) | 38 (30, 57) | |
| Male Sex | 59 (56.7) | 130 (41.5) | 166 (56.7) | 355 (50.0) | <0.001 |
| Tobacco Use | | | | | 0.330 |
| No | 55 (52.9) | 180 (57.3) | 170 (58.0) | 405 (57.0) | |
| Yes | 19 (18.3) | 39 (12.4) | 41 (14.0) | 99 (13.9) | |
| *Unanswered* | *30 (28.8)* | *95 (30.3)* | *82 (28.0)* | *207 (29.1)* | |
| Alcohol Use | | | | | 0.303 |
| No | 52 (50) | 168 (53.5) | 166 (56.7) | 386 (54.3) | |
| Yes | 21 (20.2) | 51 (16.2) | 42 (14.3) | 114 (16.0) | |
| *Unanswered* | *31 (29.8)* | *95 (30.3)* | *85 (29.0)* | *211 (29.7)* | |
| Mining status | | | | | 0.001 |
| No | 57 (54.8) | 192 (61.1) | 158 (53.9) | 407 (57.2) | |
| *Unanswered* | *41 (39.4)* | *113 (36.0)* | *105 (35.8)* | *259 (36.4)* | |
| Yes | 6 (5.8) | 9 (2.9) | 30 (10.2) | 45 (6.3) | |
| Hypertension | | | | | <0.001 |
| No | 62 (59.6) | 185 (58.9) | 171 (58.8) | 418 (58.8) | |
| Yes | 5 (4.8) | 11 (3.5) | 60 (20.5) | 76 (10.7) | |
| *Unanswered* | *37 (35.6)* | *118 (37.6)* | *62 (21.2)* | *217 (30.5)* | |
| Diabetes | | | | | 0.229 |
| No | 66 (63.5) | 194 (61.8) | 221 (75.4) | 481 (67.7) | |
| Yes | 1 (1.0) | 1 (0.3) | 6 (2.0) | 8 (1.1) | |
| *Unanswered* | *37 (35.6)* | *119 (37.9)* | *66 (22.5)* | *222 (31.2)* | |
| Epilepsy | | | | | 0.945 |
| No | 65 (62.5) | 188 (59.9) | 220 (75.1) | 473 (66.5) | |
| Yes | 2 (1.9) | 8 (2.5) | 8 (2.7) | 18 (2.5) | |
| *Unanswered* | *37 (35.6)* | *118 (37.6)* | *65 (22.2)* | *220 (30.9)* | |
| Mental illness | | | | | 1.000 |
| No | 66 (63.5) | 195 (62.1) | 225 (76.8) | 486 (68.4) | |
| Yes | 0 (0) | 1 (0.3) | 2 (0.7) | 3 (0.4) | |
| *Unanswered* | *38 (36.5)* | *118 (37.6)* | *66 (22.5)* | *222 (31.2)* | |
| Liver disease | | | | | 0.135 |
| No | 65 (62.5) | 195 (62.1) | 228 (77.8) | 488 (68.6) | |
| Yes | 1 (1.0) | 0 (0) | 0 (0) | 1 (0.1) | |
| *Unanswered* | *38 (36.5)* | *119 (37.9)* | *65 (22.2)* | *222 (31.2)* | |
| Renal insufficiency | | | | | 0.355 |
| No | 64 (61.5) | 193 (61.5) | 225 (76.8) | 488 (68.6) | |
| Yes | 1 (1.0) | 3 (1.0) | 1 (0.3) | 5 (0.7) | |
| *Unanswered* | *39 (37.5)* | *118 (37.6)* | *67 (22.9)* | *224 (31.5)* | |
| TB type | | | | | 0.050 |
| Pulmonary | 70 (67.3) | 185 (58.9) | 186 (63.5) | 441 (62.0) | |
| Extra pulmonary | 22 (21.2) | 109 (34.7) | 82 (28.0) | 213 (30.0) | |
| Both | 8 (7.7) | 12 (3.8) | 11 (3.8) | 31 (4.4) | |
| *Unanswered* | *4 (3.8)* | *8 (2.5)* | *14 (4.8)* | *26 (3.7)* | |

*(Continued)*

**Table 1.** (Continued)

|  | HIV+, no ART (N = 104) | HIV+, ART (N = 314) | HIV- (N = 293) | Total (N = 711) | P-value[1] |
|---|---|---|---|---|---|
| GeneXpert done |  |  |  |  | 0.007 |
| No | 31 (29.8) | 111 (35.4) | 69 (23.5) | 211 (29.7) |  |
| Yes | 73 (70.2) | 203 (64.6) | 223 (76.1) | 499 (70.2) |  |
| *Unanswered* | *0 (0)* | *0 (0)* | *1 (0.4)* | *1 (0.1)* |  |
| Chest x-ray done |  |  |  |  | <0.001 |
| No | 39 (37.5) | 129 (41.1) | 67 (22.9) | 235 (33.1) |  |
| Yes | 65 (62.5) | 184 (58.6) | 226 (77.1) | 475 (66.8) |  |
| *Unanswered* | *0 (0)* | *1 (0.3)* | *0 (0)* | *1 (0.1)* |  |

[1] Wilcoxon rank sum (age) and Fisher's exact test were used to test for differences in distribution by HIV/ART category. Unanswered values were excluded from testing.

recorded non-HIV comorbidity status. Of those with known comorbidity status, 15.4% had hypertension, 1.6% diabetes, 3.7% epilepsy, 0.6% mental illness, 0.2% liver disease, and 1.0% renal insufficiency.

## Down referral

Half of all patients were down referred (n = 359, 50.5%) with an additional 264 (37.1%) having a successful outcome–thus, the majority of all (87.6%) patients included in our cohort had a positive treatment outcome. Patients with HIV and on ART were more likely to be down referred (vs. unfavorable outcome) than patients with untreated HIV (1.64 OR 0.86, 3.11 95% CI) (Table 3). Individuals without HIV were much more likely to be down-referred compared to individuals with untreated HIV (2.90 OR, 1.36, 6.17 95% CI). When comparing individuals with treated HIV, individuals without HIV were also more likely to be down referred (1.77 OR, 0.93–3.35 95% CI). When imputing missing data, we found similar estimated odds ratios and 95% CIs (S2 Table in S1 File).

## Success

Among 711 patients with HIV status reported, only 12.4% (n = 88) had an unfavorable outcome, 37.5% of these were LTFU (Table 2). Individuals on ART at initiation of TB treatment were more likely to have a successful outcome compared to individuals with untreated HIV (2.29 OR, 1.13, 4.63 95% CI) (Table 3). In addition, there was an even greater likelihood for individuals without HIV having a successful TB treatment outcome compared to individuals with untreated HIV (4.98 OR, 2.07, 11.25 95% CI). When comparing individuals with treated HIV, individuals without HIV were also more likely to have a successful TB treatment outcome (2.18 OR, 1.13–4.20 95% CI). When imputing missing data, we found similar estimated odds ratios and 95% CIs (S2 Table in S1 File).

## Discussion

In this analysis, we observed high rates of positive outcomes (either success or down referral)– 88%—among all drug sensitive TB patients regardless of HIV or ART status. Compared with other studies, and South African statistics, these rates are impressive [35, 36]. South Africa estimates a nearly 54% treatment success rate among the burden of individuals with drug-sensitive TB, and a 52% treatment success rate among people co-infected with TB and HIV [35].

We found that individuals without HIV infection had nearly five times greater odds of having a successful outcome compared with individuals with untreated HIV. This disparity

**Table 2. TB treatment outcome by HIV and ART status (N = 711).**

|  | HIV+, no ART (N = 104) | HIV+, ART (N = 314) | HIV- (N = 293) | Total (N = 711) |
|---|---|---|---|---|
| **Down referred** | 56 (53.8) | 160 (51.0) | 143 (48.8) | 359 (50.5) |
| **Success** | 27 (26.0) | 116 (36.9) | 121 (41.3) | 264 (37.1) |
| Cure | 2 | 8 | 11 | 21 |
| Completed treatment | 25 | 108 | 110 | 243 |
| **Unfavorable** | 21 (20.2) | 38 (12.1) | 29 (9.9) | 88 (12.4) |
| Died | 15 | 26 | 12 | 53 |
| Treatment failure | 0 | 0 | 2 | 2 |
| Lost to follow up | 6 | 12 | 15 | 33 |

highlights the importance of integrating HIV and TB care–i.e. testing and treating HIV within TB programs and vice versa–and of paying special attention to patients with HIV who have not yet initiated ART when TB treatment is started [37]. Additionally, although we were unable to measure adherence in this study, other studies in South Africa have found that people receiving concurrent treatment for HIV/TB are highly adherent [38].

A recent study from Botswana found 88–91% of patients co-infected with HIV/TB were successfully treated for TB [39]. Dolutegravir-based ART was associated with favorable TB treatment outcomes and high rates of viral load suppression were found across all regimen

**Table 3. Estimated odds ratios for Down referral vs. Unfavorable and Successful vs. Unfavorable TB treatment outcome (N = 711).**

|  | Estimated OR for Down referral vs. Unfavorable TB Treatment Outcome | | Estimated OR for Successful vs. Unfavorable TB Treatment Outcome | |
|---|---|---|---|---|
| **Variable** | **Odds Ratio and 95% CI** | **p-value** | **Odds Ratio and 95% CI** | **p-value** |
| HIV+, no ART | 1 (Reference) | | 1 (Reference) | |
| HIV+, ART | 1.64 (0.86, 3.11) | 0.065 | 2.29 (1.13, 4.63) | 0.010 |
| HIV- | 2.90 (1.36, 6.17) | 0.003 | 4.98 (2.21, 11.25) | <0.001 |
| *Age category* | | | | |
| 15–24 | 1 (Reference) | | 1 (Reference) | |
| 25–54 | 0.73 (0.3, 1.77) | 0.244 | 0.83 (0.33, 2.08) | 0.347 |
| 55–64 | 0.53 (0.19, 1.46) | 0.108 | 0.42 (0.14, 1.21) | 0.054 |
| 65+ | 0.21 (0.08, 0.56) | 0.001 | 0.31 (0.11, 0.83) | 0.01 |
| *Sex* | | | | |
| Female | 1 (Reference) | | 1 (Reference) | |
| Male | 0.89 (0.53, 1.48) | 0.323 | 0.92 (0.54, 1.56) | 0.378 |
| *TB Type* | | | | |
| Pulmonary | 1 (Reference) | | 1 (Reference) | |
| Extra Pulmonary | 0.71 (0.4, 1.24) | 0.114 | 1.47 (0.83, 2.59) | 0.094 |
| Both | 0.34 (0.12, 0.98) | 0.023 | 1.01 (0.38, 2.74) | 0.49 |
| Unanswered | 0.57 (0.19, 1.72) | 0.158 | 0.5 (0.15, 1.73) | 0.138 |
| *Alcohol use* | | | | |
| Yes | 1 (Reference) | | 1 (Reference) | |
| No | 0.34 (0.05, 2.34) | 0.136 | 0.44 (0.06, 3.24) | 0.208 |
| Unanswered | 1.42 (0.55, 3.69) | 0.234 | 1.38 (0.51, 3.74) | 0.261 |
| *Tobacco use* | | | | |
| Yes | 1 (Reference) | | 1 (Reference) | |
| No | 1.39 (0.2, 9.77) | 0.372 | 1.02 (0.13, 7.65) | 0.494 |
| Unanswered | 0.66 (0.26, 1.69) | 0.195 | 0.49 (0.18, 1.31) | 0.078 |

categories. Another recent study from Nigeria found that 84% of patients co-infected with HIV/TB were successfully treated for TB [40]. Successful treatment was associated with being newly registered for TB, receiving TB treatment for the first time, and being treated at a private health facility. Thus, other studies have also found high rates of treatment success among patients co-infected with TB and HIV.

In an Italian cohort of 246 patients with HIV-associated TB, they found that ART initiation during TB treatment was associated with a substantial reduction of death compared to not starting ART; however, even patients already on ART when they developed TB disease remained at an elevated risk of death [41]. The data available to us in this study did not allow us to determine whether ART was initiated, nor the timing of ART initiation; however, clinicians at the hospital responsible for TB and ART patients report that the majority of patients who are naïve at the start of TB treatment remain off of ART until at least two weeks past initiation of TB treatment in order to avoid immune reconstitution inflammatory syndrome (IRIS) associated complications [42, 43]. Per conversations with treating clinicians at this rural district hospital, most patients diagnosed with HIV and TB at the same time initiate ART within two months post TB initiation, depending on their CD4 count.

As mentioned above, a down referral is considered a positive outcome. Typically, 'healthier' patients are down referred as they are seen as individuals that can be managed at the clinic level, usually by a nurse. At times, patients who are fairly unwell are also referred to their local clinic, especially when they live far away from the district hospital and transport costs to return are prohibitive, but the sickest patients are typically retained for hospital follow-up and provided with transport money by a local NGO. The decentralization of TB care has been encouraged due the preference by patients to receive care closer to home, as well as the cost-effectiveness of decentralized models of care in South Africa [44]. This cohort of TB patients had an impressive rate of down-referral, which could be seen as a positive indication of the overall health system in this setting working effectively and efficiently.

HIV continues to drive the TB epidemic in South Africa, where nearly 60% of patients are co-infected [1]. Setting policy goals and initiatives to integrate TB and HIV care is important. South Africa has been a champion of HIV care—specifically increasing HIV testing and increasing initiation of ART [5]. Due to the dual high burden of TB and HIV in the country, integrated TB/HIV care has also improved substantially in the past decade [17, 45–47]. However, more must be done for individuals suffering from dual infection, especially when ART has not yet been established. In the recent country-wide TB prevalence survey, they found that 56% of HIV positive participants had not sought care for their TB symptoms compared to 69% of individuals who were HIV negative [4]. It is also important to advocate for early and regular HIV and TB screening for all South Africans–especially men, who often have delays in seeking care and barriers to healthcare access [4]. Providing more male friendly health services are needed.

## Limitations

First, a major limitation in this cohort was the number of individuals we excluded due to missing treatment outcomes. This is always a risk when using retrospective data and routinely collected data. In our main analysis, we excluded 295 (29%) of individuals because of missing treatment outcome values or missing HIV status because we believe these are missing at random. Our supplemental analysis (S1 File) did not find any difference in demographic characteristics between the individuals excluded from the analysis, but did find that excluded individuals were slightly more likely to have extra pulmonary TB (37.6% vs. 30.0%). When we repeated our analysis with imputed missing values, we found no meaningful difference from the complete case analysis presented in the main text.

Second, we considered LTFU as an unfavorable outcome to align with existing TB research that find people who are LTFU have worse treatment outcomes [30]. Unfortunately, no information about the course of TB (nor HIV) treatment was available among patients in order to conduct a more granular time-to-event analysis, potentially accounting for informative censoring. Third, measures of adherence were not collected in the study, so we were unable to investigate the potential mediating role of adherence on the relationship between HIV/ART status and treatment outcomes [48].

Additionally, we did not have laboratory data to further investigate CD4 count and HIV viral loads to understand how each of those play a role in TB treatment outcomes [49]. We also were unable to assess timing of treatment initiation for either TB or HIV—both of which can have significant impact on treatment outcomes [6, 50]. Similarly, we do not know how many (if any) of the people living with HIV not on ART may have started ART during their TB treatment. It is likely that many individuals did start ART while engaged with care per South African guidelines; however, this was not captured longitudinally [26].

## Conclusion

Despite positive outcomes among most TB patients, HIV positive patients who are not on ART when TB treatment is initiated, do significantly worse than patients without HIV and patients with treated HIV. Therefore, patients with untreated HIV should be seen as a high-risk group, be swiftly followed up and the biopsychosocial aspects of their care need to be carefully considered. Ensuring that patients with HIV/TB coinfection are correctly treated for both diseases is paramount and the drive for the integration of TB/ART services in South Africa needs to continue.

## Supporting information

**S1 File. Supplemental analysis.** Investigating missing outcomes.
(DOCX)

## Acknowledgments

Thank you to Buyiswa Speelman and Kamila Radjabova for their help in data collection and data entry.

## Author Contributions

**Conceptualization:** Brittney J. van de Water, Suretha Cilliers, Nadishani Meyer, Michael Wilson, Catherine Young, Ben Gaunt, Karl le Roux.

**Data curation:** Brittney J. van de Water, Isabel Fulcher, Nadishani Meyer, Michael Wilson.

**Formal analysis:** Brittney J. van de Water, Isabel Fulcher.

**Funding acquisition:** Brittney J. van de Water.

**Methodology:** Brittney J. van de Water, Isabel Fulcher, Suretha Cilliers, Nadishani Meyer, Karl le Roux.

**Visualization:** Isabel Fulcher.

**Writing – original draft:** Brittney J. van de Water.

**Writing – review & editing:** Isabel Fulcher, Suretha Cilliers, Nadishani Meyer, Michael Wilson, Catherine Young, Ben Gaunt, Karl le Roux.

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
