## [Decision Letter · Decision Letter 0]

15 Sep 2021

PONE-D-21-14796Association of HIV infection and antiretroviral therapy with the occurrence of an unfavorable TB treatment outcome in a rural district hospital in Eastern Cape, South Africa: A retrospective cohort studyPLOS ONE

Dear Dr. van de Water,

Thank you for submitting your manuscript to PLOS ONE. After careful consideration, we feel that it has merit but does not fully meet PLOS ONE’s publication criteria as it currently stands. Therefore, we invite you to submit a revised version of the manuscript that addresses the points raised during the review process.

The above-referenced paper has been reviewed by two experts serving as peer reviewers for PLOS One.

 After careful review, I am sending you the manuscript for significant revisions per the reviewers' request.  One of the substantial concern from the reviewers are statistical analysis are not correctly handled. The reviewer's comments are appended below.

We look forward to receiving your revised manuscript.

Kind regards,

G. K. Balasubramani

Academic Editor

PLOS ONE

Journal Requirements:

2. In your ethics statement in the Methods section and in the online submission form, please provide additional information about the data used in your retrospective study. Specifically, please ensure that you have discussed whether all data were fully anonymized before you accessed them and/or whether the IRB or ethics committee waived the requirement for informed consent. If patients provided informed written consent to have data from their medical records used in research, please include this information.

5. Please ensure that you refer to Figure 1 in your text as, if accepted, production will need this reference to link the reader to the figure.

Additional Editor Comments (if provided):

The above-referenced paper has been reviewed by two experts serving as peer reviewers for PLOS One.

After careful review, I am sending you the manuscript for significant revisions per the reviewers' request. One of the substantial concern from the reviewers are statistical analysis are not correctly handled. The reviewer's comments are appended below.

Reviewers' comments:

Reviewer's Responses to Questions

**Comments to the Author**

1. Is the manuscript technically sound, and do the data support the conclusions?

Reviewer #1: Partly

Reviewer #2: Partly

2. Has the statistical analysis been performed appropriately and rigorously? 

Reviewer #1: No

Reviewer #2: No

3. Have the authors made all data underlying the findings in their manuscript fully available?

Reviewer #1: No

Reviewer #2: Yes

4. Is the manuscript presented in an intelligible fashion and written in standard English?

Reviewer #1: Yes

Reviewer #2: No

5. Review Comments to the Author

Reviewer #1: The authors have examined the association of HIV treatment on the TB treatment outcome. The study is based on moderate sample size data from a hospital in South Africa. The authors mention that there is not much literature on HIV treatment vs TB treatment and hence this study has some insight for treatment of TB.

There are a number of issues associated with this manuscript and the presentation has to be improved.

The main concerns are:

There is no mention about the confounding effect to type of TB which could have influence on the outcome and on HIV treatment. It is not clear if this is considered in the current analysis. Also other covariates such as alcohol and tobacco use may also have an influence on the TB outcome.

Although ORs and CIs are for the main analysis are provided, some additional details on original parameter estimates, SE and the p-values for the HIV status variable and for the covariates in the model would be useful. Results in table 3 and 4 can be provided together as these are from single multinomial logit analysis.

The authors should briefly discuss the type of TB treatment which could be useful for the reader interested in TB research.

Table 1 has lot of additional information about covariates/illness but the details about the assessment of these conditions (whether self-reported or diagnosed) are missing. Separate row for p-value in this table can be omitted. Are these reported p-values for three group or two group comparisons? It is not easy to understand this as the authors discuss the results separately for HIV treatment groups.

Some additional minor comments are:

The figure 1 which is important for the reader to understand the numbers presented in the study design is not mentioned inside the text.

Does “unanswered” refer to missing data or refused to answer (if self-reported)?

It is not clear why the authors present the age results with 936 samples when all other results are based on 711 samples.

Incidence and prevalence estimates should have a time interval.

It is hard to follow the description provided in data source and study measures. The authors should describe the information about some of the self-reported measures.

Provide the reference for R software and the r-function used for the multinomial logit model.

Reviewer #2: A major limitation of this paper is the number of people excluded in this study (414 no treatment outcome, 77 no HIV status reported and 212 < 15 years). They start with 1429 subjects with TB and the analysis sample contains only 711 (48%). They claim that missing completely was at random but no results are present to substantiate this claim.

Lines 102 -106 talks about children < 15 years that are excluded from the study and not appropriate in the methods section

Line 111 - The TB incidence rate is mentioned as 838/100,000 per year. If this is the case, then wouldn't we expect much more cases during the study period (Jan 2017 to Apr 2020)? (even after excluding children). Were all TB cases in the Eastern Cape been accounted in this study?

Line 117 Is the adult unemployment population 91.5%? This appears to be very high. (9.9% reported mining work alone!)

Line 124 Why was the rapid cartridge HIV antibody test not done for the 77 patients with no HIV status?

Line 129-130 Who created the composite categorical outcomes? Why couldn't it be created for the 414 patients?

Line 153 - Why was the "unfavorable" outcome chosen as the reference category (which includes loss to follow-up)? Why can't the majority (88%) who had positive outcomes be the reference category?

Line 155- Why can't age be introduced as a continuous variable in the models? Dichotomizing a continuous variable arbitrarily with a cut-off as 45 years may not be appropriate. Is there any clinical evidence? Would childbearing age be appropriate for males too?

Table 1: Approximately 30% of most variables in Table 1 appear to be missing (unanswered) and are excluded from tests of association. Are these variables self-reported or taken from the clinical notes?

Tables 2 and 3 are not presented in a logical manner and need to be summarized properly

Line 206 - Why are all age reported (n=936) when the study includes only >= 15 years (n=711). They are not relevant to the study

Lines 218 - 223 The statements are ambiguous and not clear. Especially lines 222 & 223 are vague.

Line 274 - It is mentioned that 341 (34%) are excluded because of missing treatment outcomes. This is not in agreement with (n=414) presented in figure 1. Figure 2 should include only the study participants and not excluded children.

6. PLOS authors have the option to publish the peer review history of their article (what does this mean?). If published, this will include your full peer review and any attached files.

Reviewer #1: No

Reviewer #2: No

---

## [Author Response · Author response to Decision Letter 0]

20 Oct 2021

RE: PLOS ONE PONE-D-21-14796 

Dear Dr. Balasubramani and reviewers,

Thank you very much for your thorough review and comments on our manuscript. We have revised the manuscript to be a much-improved version addressing all of your concerns. 

Below we list how we have addressed each reviewer’s comment. 

Ethics approval was provided by Walter Sisulu University in South Africa, the Eastern Cape Department of Health, and the Harvard Medical School Institutional Review Board.

Thank you for the opportunity to revise our manuscript. We look forward to hearing back soon.

Sincerely,

Brittney van de Water, on behalf of the authors

Editors Comments

Editor 1.1: Please ensure that your manuscript meets PLOS ONE's style requirements, including those for file naming. 

RESPONSE: Thank you. We have reviewed the style requirements and have named files appropriately. 

Editor 1.2: In your ethics statement in the Methods section and in the online submission form, please provide additional information about the data used in your retrospective study. Specifically, please ensure that you have discussed whether all data were fully anonymized before you accessed them and/or whether the IRB or ethics committee waived the requirement for informed consent. If patients provided informed written consent to have data from their medical records used in research, please include this information.

RESPONSE: Thank you. We have added into our Ethics statement “Data used in this retrospective study were identifiable upon collection; however, were fully deidentified and anonymized before statistical analysis. Both IRBs waived informed consent.”.

Editor 1.3: We note that you have indicated that data from this study are available upon request. PLOS only allows data to be available upon request if there are legal or ethical restrictions on sharing data publicly.

RESPONSE: Thank you for this guidance. We will make a limited dataset available through Boston College’s Dataverse repository. 

Editor 1.4: In your revised cover letter, please address the following prompts:

RESPONSE: Thank you. As noted above, we have no limitations and data will be deposited into Boston College’s Dataverse repository. 

b) If there are no restrictions, please upload the minimal anonymized data set necessary to replicate your study findings as either Supporting Information files or to a stable, public repository and provide us with the relevant URLs, DOIs, or accession numbers. For a list of acceptable repositories, please see: http://journals.plos.org/plosone/s/data-availability#loc-recommended-repositories. We will update your Data Availability statement on your behalf to reflect the information you provide.

RESPONSE: Many thanks.

Editor 1.4: Please include a separate caption for each figure in your manuscript.

RESPONSE: Thank you – we have added a separate caption for Figure 1 (and have omitted Figure 2 in line with reviewers’ comments).

Editor 1.5: Please ensure that you refer to Figure 1 in your text as, if accepted, production will need this reference to link the reader to the figure.

RESPONSE: Thank you, and apologies for omitting. We have included reference to Figure 1 in the first paragraph of the Methods section.

Reviewer 1 Comments

Reviewer 1.1: There is no mention about the confounding effect to type of TB which could have influence on the outcome and on HIV treatment. It is not clear if this is considered in the current analysis. Also, other covariates such as alcohol and tobacco use may also have an influence on the TB outcome.

RESPONSE: Thank you for this comment. We have now adjusted for TB type, alcohol use, and tobacco use in the analysis with minimal change, as detailed in the Methods, Statistical Analysis section.

Reviewer 1.2: Although ORs and CIs are for the main analysis are provided, some additional details on original parameter estimates, SE and the p-values for the HIV status variable and for the covariates in the model would be useful. Results in table 3 and 4 can be provided together as these are from single multinomial logit analysis.

RESPONSE: Thank you. We have added p-values to tables 3 and 4 and have combined them into just one table, Table 3. Note that we do not report standard errors as these are less interpretable on the odds ratio scale. 

Reviewer 1.3: The authors should briefly discuss the type of TB treatment which could be useful for the reader interested in TB research.

RESPONSE: Thank you. We have added “The vast majority of adults > 15 years with drug sensitive TB in South Africa is managed with four drugs for the first two months of the “intensive phase“ of treatment: rifampicin, isoniazid, pyrazinamide and ethambutol (RHZE), followed by four months of rifampicin and isoniazid (RH) – the “continuation phase”, as per the South African National TB guidelines.“ into the Methods section for background on types of TB treatments used during this period in South Africa.

Reviewer 1.4: Table 1 has lot of additional information about covariates/illness but the details about the assessment of these conditions (whether self-reported or diagnosed) are missing. Separate row for p-value in this table can be omitted. Are these reported p-values for three group or two group comparisons? It is not easy to understand this as the authors discuss the results separately for HIV treatment groups.

RESPONSE: Thank you for this comment. We added in the Methods, Data sources and study measures section that “Baseline demographic and medical history were collected through self-report and an interview”. To clarify, we have added into the Methods that p-values correspond to a Fisher’s exact test that compares across all three groups simultaneously (HIV negative, HIV positive on ART, and HIV not on ART). This can be interpreted as testing for differences in the distribution of HIV/ART for each demographic characteristic.

Reviewer 1.5: Figure 1 is important for the reader to understand the numbers presented in the study design is not mentioned inside the text.

RESPONSE: Thank you for noticing this. We have now referenced Figure 1 in the text at first mention in the Methods section, Study design and study population.

Reviewer 1.6: Does “unanswered” refer to missing data or refused to answer (if self-reported)?

RESPONSE: Thank you for this question. As described in the Methods, “unanswered” refers to questions that were typically not asked by the clinician. In trying to not “patient blame” we want readers to understand that often it is not because patients are non-forthcoming with their health status, rather often times it is the provider (nurse or doctor) who never asks the question and/or leaves it blank. Of note, these charts are from a low-resourced setting where clinicians are often hard pressed for time and charting is not always the priority. The sentence reads “Missing values for covariates were given their own category of “Missing”, often when a clinician did not ask or did not tick “yes” or “no” in the medical chart”.

Reviewer 1.7: It is not clear why the authors present the age results with 936 samples when all other results are based on 711 samples.

RESPONSE: Thank you for this comment. Our original analysis only focused on adults, but we chose to display results for individuals <15 years in Figure 2. We now realize that this caused confusion and have removed persons <15 years from Figure 2 and only focus on adults in this manuscript and Figure 2. 

Reviewer 1.8: Incidence and prevalence estimates should have a time interval.

RESPONSE: Thank you for this concern. We have added dates for prevalence and incidence referenced in the Methods per the cited references.

Reviewer 1.9: It is hard to follow the description provided in data source and study measures. The authors should describe the information about some of the self-reported measures.

RESPONSE: Thank you for this comment. We have revised the Methods section which now reads “Baseline demographic and medical history were collected through self-report and an interview with a healthcare provider as the TB file was started…” and expands on self-reported measures. 

Reviewer 1.10: Provide the reference for R software and the r-function used for the multinomial logit model.

RESPONSE: Thank you for catching this. We have now included the R software and R-function used for analysis in the Methods. We also include the R scripts necessary to run our analyses. 

Reviewer 2 Comments

Reviewer 2.1: A major limitation of this paper is the number of people excluded in this study (414 no treatment outcome, 77 no HIV status reported and 212 < 15 years). They start with 1429 subjects with TB and the analysis sample contains only 711 (48%). They claim that missing completely was at random but no results are present to substantiate this claim.

RESPONSE: We appreciate your concern. We have revised the Methods to only describe individuals who were > 15 years. Therefore, it now reads “Data for this study were taken from TB medical charts of a cohort of individuals 15 years or older who were diagnosed with drug-sensitive TB and who should have completed treatment at the district hospital between January 4, 2017 and April 27, 2020 (N=1006) (Fig 1). There were 295 individuals excluded due to no documented treatment outcome (n=246) or no HIV status reported (n=49) for a final sample size of 711.” We also added into the Data Sources/Study Measures section “Missing values for covariates were given their own category of “Missing”, often when a clinician did not ask or did not tick “yes” or “no” in the medical chart, as described above, this is a busy hospital with busy clinicians and lacks administrative capacity to close loops for charting, which does not impact clinical care. For missing HIV status, often it was due to tests not being available on the day a patient was seen, and then no HIV status was recorded upon testing on a subsequent day after the initial baseline data collection. Finally, a final treatment outcome is not always recorded by clinicians in the medical chart. Therefore, we believe these missing data points are missing at random.”. Also, we have now included a supplemental analysis where we (1) investigate differences in demographic characteristics between our included study population and those that were excluded due to missingness and (2) repeated analysis with a multiple imputation procedure to impute the missing outcomes and HIV status. Importantly, we feel confident that our results were not impacted by this large source of missingness as we found minimal differences in demographic characteristics and the results did not change in our imputed version. 

Reviewer 2.2: Lines 102 -106 talks about children < 15 years that are excluded from the study and not appropriate in the methods section.

RESPONSE: Thank you. We have removed the sentences regarding the rationale and only provide inclusion/exclusion criteria now. We have also cited 3 additional sources explaining why we excluded individuals < 15 years. 

Reviewer 2.3: Line 111 - The TB incidence rate is mentioned as 838/100,000 per year. If this is the case, then wouldn't we expect much more cases during the study period (Jan 2017 to Apr 2020)? (even after excluding children). Were all TB cases in the Eastern Cape accounted in this study?

RESPONSE: Thank you for this observation. No, not all cases of TB were accounted for in this study – this study was only at one district hospital in a rural area with a population of nearly 130,000. There are 12 referring clinics to the hospital where many cases of TB are also diagnosed. We only provided this information as background information about the setting.

Reviewer 2.4: Line 117 Is the adult unemployment population 91.5%? This appears to be very high. (9.9% reported mining work alone!)

RESPONSE: Thank you. Reviewing the data we took from the official 2011 South African Census (as presented through Wazimap.co.za), we realize that the manner in which we have reported this data may be unclear. Although the rate of employment in the catchment area of Zithulele Hospital, at 9.7% of working-age adults is accurate and indeed exceedingly low, the remainder of the adults should not, strictly speaking, be categorized as “unemployed”. The South African Census data uses the following categories and definitions: 

Employed: Persons who worked for pay, profit, or family gain, even for just one hour, in the seven days before they were interviewed.

Unemployed (official definition): Persons who did not work, but who looked for work and were available to work in the past seven days.

Discouraged work-seeker: Someone who has lost hope in finding any kind of work.

Not economically active: Persons who were neither employed nor unemployed (e.g. full-time students; retired persons; and homemakers who did not want to work).

Looking at the areas/wards making up the catchment area of Zithulele Hospital (excluding Mthatha, a major city in KSD sub-district), the percentages for each category are as follows:

Employed: 9.7%

Unemployed: 9.6%

Discouraged work seeker: 10.5%

Not economically active: 70.2%

It is therefore more accurate to report an employment rate of 9.7%, an unemployment rate of 20% (of which half are discouraged work seekers) and an extremely high proportion (70.2%) of economically inactive people - which we have now corrected in the manuscript. 

We have also changed the Setting section to now read “The median annual household income is R14600 (US$1000)”.

Reviewer 2.5: Line 124 Why was the rapid cartridge HIV antibody test not done for the 77 patients with no HIV status?

RESPONSE: Thank you for this astute observation. As with all types of clinical care in low resource settings, not all evaluations and tests are done for every individual. At times the hospital runs out of cartridges, clinicians are too rushed to do an examination at the initial visit and plan to do it later, or in most cases the rapid test was completed, and just not documented in the source document. Three co-authors are clinicians in the hospital and confirm that due to extenuating circumstances, things do get missed from time to time. 

Reviewer 2.6: Line 129-130 Who created the composite categorical outcomes? Why couldn't it be created for the 414 patients?

RESPONSE: Thank you. We (BvdW, IF, and KlR) created the composite categorical outcomes. These were created from source document outcomes including: loss to follow-up, death, treatment failure, cure, completed treatment, and down referral. Those 414 individuals did not have a source document outcome documented. Similar to the above response regarding missing HIV status, clinicians do not always document a final outcome in the TB Medical Record. We have now included a Supplement with an investigation into the missing data. Please see our response to Comment 2.1.

Reviewer 2.7: Line 153 - Why was the "unfavorable" outcome chosen as the reference category (which includes loss to follow-up)? Why can't the majority (88%) who had positive outcomes be the reference category?

RESPONSE: Thank you. We chose “unfavorable” as the reference category because both “success” and “down referral” are considered “positive” outcomes, so it did not make sense to do a comparison of “success” to “down referral” but rather compare each of these to “unfavorable”. Importantly, this was the reference category preferred by our collaborators at the hospital, and we agree that it is easier to interpret “positive” outcomes vs. “negative” outcomes in this manner. Lastly, the reference category does not change the results and an interested reader has all the available information to “flip” the reference category if they so wished. 

Reviewer 2.8: Line 155- Why can't age be introduced as a continuous variable in the models? Dichotomizing a continuous variable arbitrarily with a cut-off as 45 years may not be appropriate. Is there any clinical evidence? Would childbearing age be appropriate for males too?

RESPONSE: Thank you for this question. We chose not to include age as a continuous (linear) term because this would force the relationship between age and treatment outcome to be linear, which it is not. However, we agree with the Reviewer that this binary cutoff at 45 years is likely not sufficient. Our remaining options were to model age: (1) using a spline or polynomial terms or (2) with more categories informed from prior literature or our data. We chose the latter as this is more common in the literature, and we believe it accurately captures the non-linear relationship between age and the treatment outcome and is similar to what the recent South African Prevalence Survey had with age brackets. (Figure 2).

Reviewer 2.9: Table 1: Approximately 30% of variables in Table 1 appear to be missing (unanswered) and are excluded from tests of association. Are these variables self-reported or taken from the clinical notes?

RESPONSE: Thank you. Yes, we agree that many of the variables have high rates of missingness, hence, why we reported the amount “unanswered” in the table for transparency. This study was a retrospective chart review, and therefore all variables were taken from clinical charts (where they are primarily self-reported as described in the Methods). 

Reviewer 2.10: Tables 2 and 3 are not presented in a logical manner and need to be summarized properly.

RESPONSE: Thank you for this helpful comment. We have combined Tables 3 and 4 into one Table, per other Reviewer comments, and additionally, to address your concern here, we have rearranged our Results to discuss down referral prior to successful outcomes. 

Reviewer 2.11: Line 206 - Why are all age reported (n=936) when the study includes only >= 15 years (n=711). They are not relevant to the study.

RESPONSE: Thank you. We have omitted this paragraph to help streamline the paper and now only focus on adults (along with Figure 2).

Reviewer 2.12: Lines 218 - 223 The statements are ambiguous and not clear. Especially lines 222 & 223 are vague.

RESPONSE: Thank you for this comment. We have deleted those three sentences for clarity. 

Reviewer 2.13: Line 274 - It is mentioned that 341 (34%) are excluded because of missing treatment outcomes. This is not in agreement with (n=414) presented in figure 1. 

RESPONSE: Thank you. Apologies, it was a mistake that we mentioned n=341. We have edited the Methods to say “Data for this study were taken from TB medical charts of a cohort of individuals 15 years or older who were diagnosed with drug-sensitive TB and who should have completed treatment at the district hospital between January 4, 2017 and April 27, 2020 (N=1006) (Fig 1). There were 295 individuals excluded due to no documented treatment outcome (n=246) or no HIV status reported (n=49) for a final sample size of 711.” 

Reviewer 2.14: Figure 2 should include only the study participants and not excluded children.

RESPONSE: Thank you – we have revised Figure 2.

---

## [Decision Letter · Decision Letter 1]

15 Mar 2022

Association of HIV infection and antiretroviral therapy with the occurrence of an unfavorable TB treatment outcome in a rural district hospital in Eastern Cape, South Africa: A retrospective cohort study

PONE-D-21-14796R1

Dear Dr. van de Water,

We’re pleased to inform you that your manuscript has been judged scientifically suitable for publication and will be formally accepted for publication once it meets all outstanding technical requirements.

Kind regards,

Petros Isaakidis MD, PhD

Academic Editor

PLOS ONE

Reviewers' comments:

Reviewer's Responses to Questions

**Comments to the Author**

1. If the authors have adequately addressed your comments raised in a previous round of review and you feel that this manuscript is now acceptable for publication, you may indicate that here to bypass the “Comments to the Author” section, enter your conflict of interest statement in the “Confidential to Editor” section, and submit your "Accept" recommendation.

Reviewer #1: All comments have been addressed

2. Is the manuscript technically sound, and do the data support the conclusions?

Reviewer #1: Yes

3. Has the statistical analysis been performed appropriately and rigorously? 

Reviewer #1: Yes

4. Have the authors made all data underlying the findings in their manuscript fully available?

Reviewer #1: Yes

5. Is the manuscript presented in an intelligible fashion and written in standard English?

Reviewer #1: Yes

6. Review Comments to the Author

Reviewer #1: The authors have substantially revised the manuscript. Some minor points.

It is important to acknowledge the authors of the R packages. Please include references for R packages.

Table 2 could be mentioned before table 3.

7. PLOS authors have the option to publish the peer review history of their article (what does this mean?). If published, this will include your full peer review and any attached files.

Reviewer #1: **Yes: **Anbupalam Thalamuthu

---

## [Editor Report · Acceptance letter]

23 Mar 2022

PONE-D-21-14796R1 

Association of HIV infection and antiretroviral therapy with the occurrence of an unfavorable TB treatment outcome in a rural district hospital in Eastern Cape, South Africa: A retrospective cohort study 

Dear Dr. van de Water:

I'm pleased to inform you that your manuscript has been deemed suitable for publication in PLOS ONE. Congratulations! Your manuscript is now with our production department. 

Kind regards, 

on behalf of

Dr. Petros Isaakidis 

Academic Editor

PLOS ONE